

# Nutritional properties of selected superfood extracts and their potential health benefits

Jacqueline P. Barsby[1,2], James M. Cowley[1], Shalem Y. Leemaqz[2,3], Jessica A. Grieger[2,4], Daniel R. McKeating[5], Anthony V. Perkins[5], Susan E. P. Bastian[1], Rachel A. Burton[1] and Tina Bianco-Miotto[1,2]

[1] Waite Research Institute and School of Agriculture, Food and Wine, University of Adelaide, Adelaide, SA, Australia
[2] Robinson Research Institute, University of Adelaide, Adelaide, SA, Australia
[3] College of Medicine and Public Health, Flinders University of South Australia, Bedford Park, SA, Australia
[4] Adelaide Medical School, University of Adelaide, Adelaide, SA, Australia
[5] School of Medical Science, Griffith University, Southport, QLD, Australia

Corresponding author
Tina Bianco-Miotto,
tina.bianco@adelaide.edu.au

## ABSTRACT

**Background:** The term 'superfoods' is used to market foods considered to have significant health benefits. 'Superfoods' are claimed to prevent diseases as well as improving overall health, though the lack of explicit criteria means that any food can be labelled 'super' without support from scientific research. Typically, these 'superfoods' are rich in a particular nutrient for example antioxidants or omega-3 fatty acids. The objective of this study was to investigate the nutritional properties of a selection of superfood seeds: flax, chia, hulled sunflower and two types of processed hemp seeds and determine whether they may have potential health benefits.

**Methods:** We developed a simple aqueous extraction method for ground seeds and analysed their composition by mineral, protein and monosaccharide analyses. Cell viability assays were performed on Caco-2 and IEC-6 intestinal epithelial cells using increasing doses of the prepared extracts.

**Results:** Increased cell viability was observed in both cell lines with increasing concentrations of the flax seed, chia seed or hulled sunflower extracts ($P < 0.05$). Compositional analyses revealed the presence of polysaccharides, proteins and essential minerals in the aqueous extracts and *in vitro* assays showed sunflower had the highest antioxidant activity. However, differences in extract composition and antioxidant properties could not be directly related to the observed increase in cell viability suggesting that other components in the extracts may be responsible. Future studies will further characterize these extracts and investigate whether they are beneficial for gastrointestinal health.

## INTRODUCTION

Nearly 40 years ago, functional foods were idealized as a concept in which foods could beneficially affect one or more targeted functions in the body, beyond adequate nutritional effects (*Scientific Concepts of Functional Foods in Europe, 1999*). Although different definitions exist, a common characteristic is that, compared with regular foods, functional foods have beneficial physiological effects and/or reduce the risk for disease development due to the addition, or removal of certain nutrients (*Arai, 2002*). More recently, the term "superfoods" has been introduced to describe foods with specific health benefits. They are widely reported on in the media, but with no generally accepted definition, any food can be labelled as such (*Proestos, 2018*). Foods including flax, hemp, chia and sunflower seeds (botanically hemp, chia and sunflower seeds are nutlets), have all been described as superfoods (*van den Driessche, Plat & Mensink, 2018*) however, there is a paucity of literature on their health benefits.

Flax (*Linum usitatissimum*) seeds and chia (*Salvia hispanica*) seeds have high levels of omega-3 fatty acids (*Goyal et al., 2014*; *Ixtaina et al., 2011*), whereas hemp (*Cannabis sativa*) and sunflower (*Helianthus annuus*) seeds have greater amounts of omega-6 fatty acids (*Callaway, 2004*). These superfoods are claimed, by the media, to have a large range of health benefits, but they are typically not supported by scientific evidence (*Oude Groeniger et al., 2017*). Like other seeds, hemp seed foods are claimed to be good sources of protein, dietary fibre and omega-3 fatty acids. However, the sale of hemp seed foods has historically been prohibited in Australia due to legal issues concerning the psychoactivity of tetrahydrocannabinol (THC) in some strains of Cannabis. Subsequent changes to the Australia New Zealand Food Standards Code (Food Standards Australia New Zealand) in November 2017 legally permitted sale of foods containing industrial hemp ingredients with THC levels of less than 1%, to the general public.

In pre-clinical studies, flax seed reduced the level of intracellular reactive oxygen species and protected genomic DNA against damage in V79 Chinese hamster pulmonary fibroblasts (*Skorkowska-Telichowska et al., 2016*). In diet-induced obese rats, chia seed oil demonstrated anti-inflammatory effects which reduced the severity of rheumatoid arthritis (*Mohamed, Mohamed & Fouda, 2020*). Ingested chia seed and chia seed oil have been shown to improve systemic glucose and insulin tolerance in rats (*Marineli Rda et al., 2015*). A recent review of sunflower seeds, summarised the content of nutrients, minerals, antioxidants and vitamins, and their roles in antioxidant, antimicrobial, antidiabetic, antihypertensive, anti-inflammatory and wound-healing effects (*Guo, Ge & Na Jom, 2017*; *van den Driessche, Plat & Mensink, 2018*). There has been limited literature on hemp seeds in food products, but the field is rapidly expanding and promising, with current research showing beneficial effects on human health (*Farinon et al., 2020*).

Previous *in vitro* extraction methods for seed constituents have typically used very high temperatures and harsh solvents like ethanol and methanol (*Butt et al., 2019*). Although these methods yield a high concentration of bioactive compounds, they may not necessarily replicate what can be extracted by the human body after ingestion of these superfoods, and may heavily over-represent cytotoxic compounds in the extract that are

typically poorly-accessible to the digestive system (*Alminger et al., 2014*; *Bohn et al., 2018*; *Minekus et al., 2014*). By creating a new extraction method without the use of harsh solvents or high temperatures we can eliminate some of the cytotoxic compounds and also have a better replication of what the human body would be able to extract and use.

The objective of this study was to develop a mild aqueous extraction method, performed at human body temperature, which may better mimic what could be extracted from 'superfood' seeds when ingested. This extraction generated the products flax seed extract (FSE), chia seed extract (CSE), hulled sunflower extract (HSE), hulled hemp extract where just the hemp hearts are used (HHE) and hemp flour extract (HFE). This study investigated the nutrient composition of these extracts, and the impact on cell growth by *in vitro* cell viability assays on IEC-6 and Caco-2 intestinal epithelial cells, the most commonly used cell lines for testing novel food products (*Sambruy et al., 2001*). In addition, we profiled the nutritional composition of these seed extracts and the flours they were derived from.

## MATERIALS & METHODS

### Materials

Chia (Organic Road, Australia), flax (Lotus, Australia), hulled sunflower seeds (Natural Road, Australia) and hulled hemp seeds (Hemp Foods Australia, Australia) were purchased commercially. A pre-ground and processed commercial hemp flour (Hemp Foods Australia) was also purchased, this flour was made from whole hemp seed with the oil pressed out and some of the protein extracted.

### Flour preparation

Approximately one cup of seeds were ground using a blade mill-type Multi Grinder (Sunbeam, Australia) for approximately 5 min at room temperature to create a seed meal. The meal was sieved and larger pieces were ground again. This flour was sealed in a falcon tube and placed in a sealed container in the fridge with silica beads to prevent excess moisture.

### Seed and flour nutritional profile

Where relevant, nutritional information such as protein, energy (kJ), fat, dietary fibre and carbohydrate contents were taken from the commercial packaging of the purchased seeds. Lipid content and fatty acid profiles of flours were determined by the Waite Lipid Analysis Service, at the South Australian Health and Medical Research Institute (*Liu, Mühlhäusler & Gibson, 2014*). Total lipid in whole seed flour was determined by modified (*Folch, Lees & Sloane Stanley, 1957*) and fatty acid profiles were determined by gas chromatography of transesterified lipids following (*Liu, Mühlhäusler & Gibson, 2014*).

### Crude aqueous body temperature seed extracts

To generate a crude aqueous body temperature extract, 4 g of flour was added to 200 mL of RO (reverse osmosis) water at 37 °C in a borosilicate glass beaker and the mixture was left to agitate on a heated magnetic stirrer to maintain this temperature. After 3 h, the

mixture was transferred into 50 mL polypropylene tubes and centrifuged at 1,000 rpm ($200 \times g$) for 10 min to pellet any debris. The supernatant was removed and freeze dried to a constant weight using a Freezone six freeze dry system (Labconco, Kansas City, MO, USA). Yield of extract was determined as the percentage of freeze-dried extract mass to mass of flour (Figure 1). Freeze dried extracts were dissolved in sterile PBS (Life Technologies™, Carlsbad, CA, USA) at a concentration of 5 mg/mL for cell culture experiments and sterile RO water for compositional analysis.

## IEC-6 and Caco-2 cell culture

IEC-6 (CRL1592) and Caco-2 (HTB-37) cells originally purchased from the ATCC were provided by SEPB and cultured in Gibco® Dulbecco's Modified Eagle Medium (DMEM; Life Technologies™, Carlsbad, CA, USA) with 10% Fetal Bovine Serum (FBS; Sigma, St. Louis, MO, USA), antibiotic/antimycotic solution (10,000 units penicillin, 10 mg streptomycin and 25 µg amphotericin B per mL) (Sigma, St. Louis, MO, USA). The cells were maintained in a 37 °C incubator with 5% $CO_2$ in 75 cm² flasks (Greiner bio-one, Kremsmünster, Austria) and the media was changed every 72 h. Cells were passaged at 70–80% confluency and split at a 1:3 ratio using Trypsin (Gibco®, Amarillo, TX, USA) to dissociate the cells from the flask surface. Cells were stained with trypan blue (Gibco®, Amarillo, TX, USA) and counted using a haemocytometer on an Olympus inverted microscope.

## Cell viability assay

A dose-response cell viability assay was performed as per *Kumar, Nagarajan & Uchil (2018)*. IEC-6 or Caco-2 cells (*Sambruy et al., 2001*) were plated in 96 well plates at a concentration of $1 \times 10^5$ cells/well in 50 µL of DMEM with 10% FBS, then left for 24 h to adhere. Extracts (5 mg/mL in PBS) were diluted in DMEM with 10% FBS to 1 mg/mL and then two-fold serial dilutions were prepared (resulting in concentrations of 1 mg/mL, 0.5 mg/mL, 0.25 mg/mL, 0.125 mg/mL, 0.0625 mg/mL and a vehicle control, 0 mg/mL). The extract treatments (50 µL) were applied to triplicate wells, and the plate was left to incubate for another 24 h. MTT solution (Invitrogen, Waltham, MA, USA) was diluted with PBS to 1 mg/mL, filter sterilized and 50 µL was added to every well using a multichannel pipette and the plate was incubated for 4 h at room temperature. A 100 µL volume of dimethyl sulfoxide (DMSO; Sigma-Aldrich, St. Louis, MO, USA) was added to every well, the plate was covered in aluminium foil and agitated on a plate shaker for 15 min to extract the formazan product. Absorbance was read using a Thermo Multiskan Spectrum spectrophotometer at 570 nm. The results were converted into a percentage by comparing them to the vehicle control value:

$$Cell\ viability\ (\%) = \frac{abs570\ (sample)}{abs570\ (control)} * 100$$

Each cell viability assay was repeated to obtain 3–4 independent experiments per cell line.
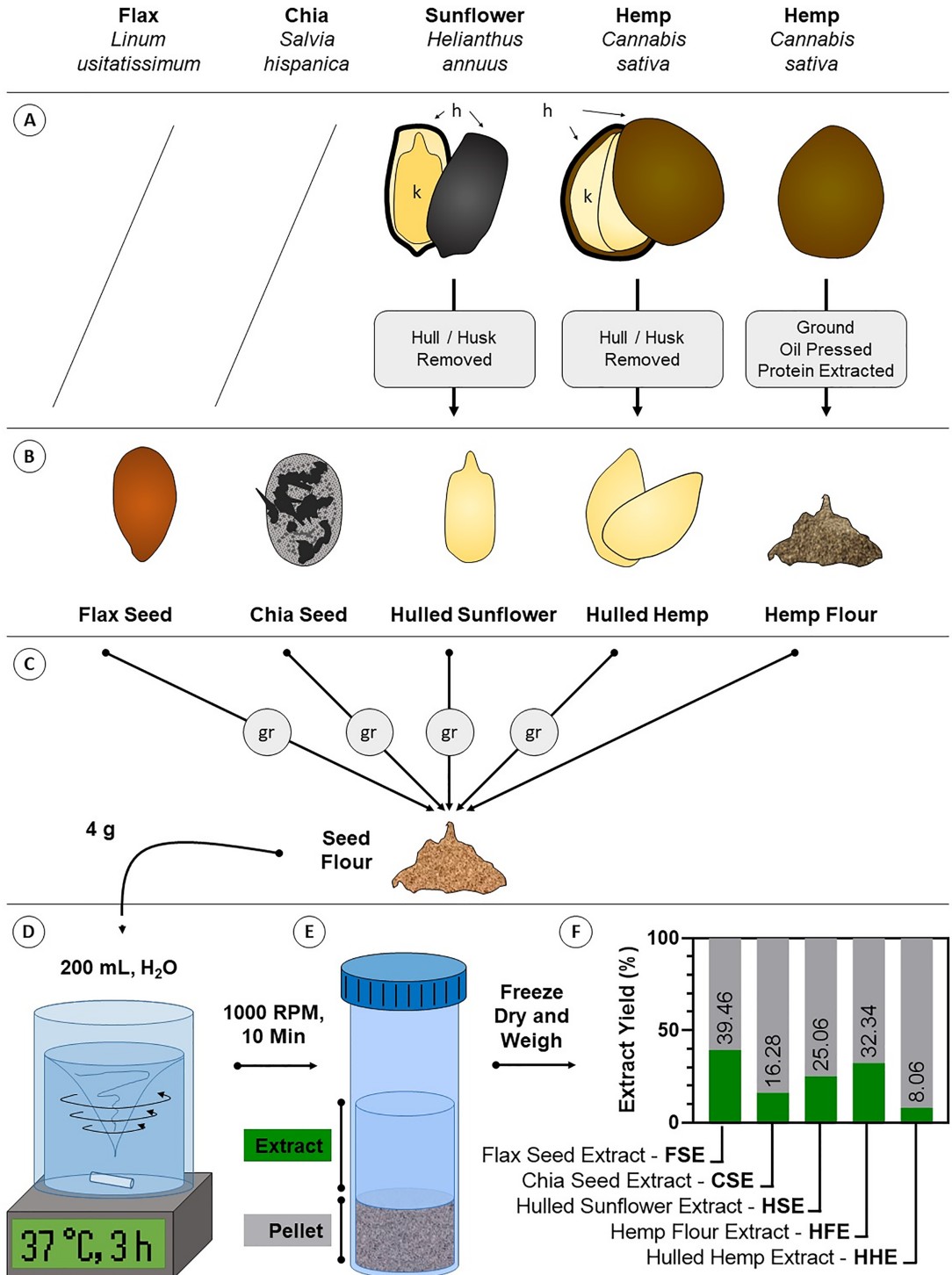

**Figure 1 Diagram of crude aqueous extract preparation.** (A) Some seeds require processing to remove inedible components like the husk. Flax and chia seeds are consumed unprocessed, while sunflower and hemp seeds are commonly consumed as kernels (k) where the indigestible hull/husk (h) is removed (hulled sunflower and hulled hemp). Hemp is also consumed as a flour where oil and crude protein have been extracted for other products (hemp flour). (B) These forms of the seeds are the most commonly consumed and were the starting material for extraction. (C) Except hemp flour, seeds were ground (gr) to flour. (D) Four g of seed flour was extracted in body temperature (37 °C) water for 3 h with intense agitation. (E) Debris was pelleted by centrifugation and the aqueous extracts were decanted and dried. (F) Dried extracts were weighed and compared against the starting material to determine the aqueous extract yield.
## Extract characterisation

### Protein content

A Qubit Fluorometric Protein Assay Kit (Invitrogen, St. Louis, MO, USA) was used to determine the protein content of the crude aqueous extracts (1 mg/mL), in triplicate, as per the manufacturer's instructions.

### Monosaccharide analysis

Monosaccharide analysis was performed in duplicate and determined using reverse phase high performance liquid chromatography (RP-HPLC) of 1-phenyl-3-methyl-5-pyrazoline (PMP) derivatives prepared following *Hassan et al. (2017)* with modifications listed in *Cowley et al. (2020)*. Quantification was performed using the area under the peaks compared to standard curves of D-mannose, D-ribose, L-rhamnose, D-glucuronic acid, D-galacturonic acid, D-glucose, D-galactose, D-xylose, L-arabinose and L-fucose.

### Mineral content

Ultra-trace element analysis of freeze-dried extracts and their source flour was determined by inductively coupled plasma-mass spectrometry (ICP-MS) following (*Hofstee et al., 2019*; *McKeating et al., 2020*) as a fee-for-service by the Perkins Pregnancy Research Laboratory (Griffith University, Brisbane, Australia). All runs were spiked with [45]Sc, [89]Y, [115]In, [159]Tb internal standards to ensure recovery and a calibration standard set was run before sample analysis. A spiked quality control run was performed after every six samples followed by a wash step. Data presented are the average of two technical repeats with minimal error. Mineral extractability was estimated using the following equation:

$$\text{Extractability } (\%) = \frac{(C^F \times M^F)}{(C^E \times M^E)} \times 100$$

where $C^F$ and $C^E$ refer to the mineral concentration in mg/kg in source flour and extract, respectively, and $M^F$ and $M^E$ refer to the mass in grams of flour used for extraction and recovered extract respectively.

## Ferric reducing antioxidant power (FRAP) assay

A FRAP assay was performed in triplicate as per Benzie and Strain (*Benzie & Strain, 1996*). Each of the crude aqueous seed extracts were serially diluted using dimethyl sulfoxide (DMSO; Sigma-Aldrich, St. Louis, MO, USA) at 1:2 dilution, eight times, with a starting concentration of 1 mg/mL. The ascorbic acid control was also diluted in the same manner. A ferrous sulphate solution was diluted using milli-Q water at concentrations of 0, 0.1, 0.2, 0.4, 0.5, 0.6, 0.8 and one mM. A 15 μL aliquot of each of the extract samples were added at multiple concentrations in a 96 well plate. A 15 μL volume of the FRAP reagent containing acetate buffer, TPTZ (2,4,6-Tris(2-pyridyl)-s-triazine) and $FeCl_3$ (iron (III) chloride) were added to every well and the results were read on a spectrophotometer at 570 nm.
## Statistical analysis

Where applicable, statistical differences in extract composition were determined by ANOVA with a post-hoc Tukey's Honestly Significant Difference (HSD) test in GenStat 15[th] Edition.

For MTT assays, marginal models for viability were fitted using Generalized Estimating Equations using R version 3.6.3 (R Foundation for Statistical Computing, Vienna, Austria), accounting for repeated experiments assuming an independence working correlation structure. Cubic splines were used to explore the non-linear dose-response relationship of concentration on viability. Separate marginal models were fitted with categorical concentrations to allow for comparisons to the DMEM + FBS (vehicle) group.

## RESULTS

### Fatty acid and micronutrient analyses of ground flours

Prior to preparing extracts for cell treatment, the seeds were ground into a flour. Tests on the flour indicated that they all contain beneficial and essential micronutrients including magnesium, calcium, iron and zinc (Table 1). Flax seed flour had the highest content of omega-3 fatty acids in comparison to omega-6 at a ratio of 11:3, followed by chia flour at 3:1 (Table 1). Comparatively, hulled hemp seed flour, processed hemp flour and sunflower seed flour all had higher levels of omega-6 compared to omega-3; sunflower flour had the highest ratio (1:614 of omega-3:omega-6). Chia seeds showed a significantly higher amount of carbohydrates (37%) in comparison to the other seeds (Table 1). Hemp flour had the highest level of dietary fibre (67%), followed by chia (35%) then flax (29%). Sunflower had the highest total fat content (57%), followed by hemp seed (54%), flax (43%), and chia (29%), with hemp flour being the lowest (9%) (Table 1).

All seeds contained magnesium with hulled hemp seeds having the highest level (5,359 mg/kg), followed by flax (4,425 mg/kg), while chia, sunflower and hemp flour had slightly lower levels (3,296–3,525 mg/kg). Both the hulled hemp seed and hemp flour contained the highest levels of iron (196 mg/kg) compared to the other seeds (80–145 mg/kg). Hulled hemp seed had the highest level of zinc, followed by hulled sunflower, hemp flour and chia, with flax having the lowest level of zinc (Table 1). Only flax seed had detectable levels of sodium at 1,073 mg/kg compared to the other seeds. Chromium, cobalt, selenium and molybdenum were present in very low levels or not detectable.

### Aqueous seed extract composition

To produce material that could be used for treating the cells, aqueous extracts FSE, CSE, HSE, HHE and HFE were generated from flax seed, chia seed, hulled sunflower, hulled hemp and hemp flour, respectively (Fig. 1). The same method was used on each seed yet varying amounts of extractable material were produced. Yield was highest for FSE at 39.46%, followed by HHE at 32.34%, HSE at 25.06%, CSE at 16.28% and lastly HFE at 8.06%. These numbers were calculated by comparing the initial weight (4 g) with the freeze-dried weight of the aqueous extract.
**Table 1 Seed flour nutrient composition with information provided on the packaging by commercial manufacturer and from independent testing.**

| | | Flax seed | Chia seed | Sunflower seed | Hulled hemp seed | Hemp flour |
|---|---|---|---|---|---|---|
| | | *Linum usitatissimum* | *Salvia hispanica* | *Helianthus annuus* | *Cannabis sativa* | |
| **Seed nutrient composition provided on packaging (per 100 g)** | | | | | | |
| Energy (kJ/100 g) | | 2,172 | 2,100 | 2,755 | 2,599 | 1,680 |
| Protein (%) | | 21 | 21 | 21 | 31.3 | 17 |
| Carbohydrate (%) | | <1 | 37 | 7.2 | 2 | 6.7 |
| Dietary fibre (%) | | 28.8 | 35 | 8.8 | 3.3 | 67.1 |
| Fat (%) | | 43 | 29 | 57 | 54.2 | 8.9 |
| **Fatty acid composition of seed oil** | | | | | | |
| **Saturated fatty acids** | Total (%) | 10.8 | 11.8 | 11.9 | 10.8 | 12.7 |
| Myristic acid (14:0) (%) | | 0.0 | 0.0 | 0.1 | 0.1 | 0.0 |
| Pentadecylic acid (15:0) (%) | | 0.0 | 0.0 | 0.0 | 0.0 | 0.0 |
| Palmitic acid (16:0) (%) | | 5.9 | 7.6 | 6.8 | 6.3 | 8.0 |
| Margaric acid (17:0) (%) | | 0.1 | 0.1 | 0.0 | 0.1 | 0.1 |
| Stearic acid (18:0) (%) | | 4.3 | 3.5 | 3.8 | 3.3 | 3.3 |
| Arachidic acid (20:0) (%) | | 0.2 | 0.3 | 0.3 | 0.7 | 0.8 |
| Behenic acid (22:0) (%) | | 0.1 | 0.1 | 0.7 | 0.2 | 0.3 |
| Lignoceric acid (24:0) (%) | | 0.1 | 0.1 | 0.3 | 0.1 | 0.2 |
| **Unsaturated fatty acids** | Total (%) | 89.2 | 88.2 | 88.1 | 89.2 | 87.3 |
| **ω-3** | Total (%) | 56.4 | 61.3 | 0.1 | 14.5 | 15.7 |
| Alpha-Linolenic acid (18:3 n − 3) (%) | | 56.4 | 61.3 | 0.1 | 14.5 | 15.7 |
| **ω-6** | Total (%) | 14.7 | 19.9 | 61.4 | 58.4 | 57.7 |
| Linoleic acid (18:2 n − 6) (%) | | 14.7 | 19.8 | 61.4 | 57.9 | 57.3 |
| Eicosadienoic Acid (20:2 n − 6) (%) | | 0.0 | 0.1 | 0.0 | 0.1 | 0.0 |
| **ω-7** | Total (%) | 1.0 | 0.9 | 0.8 | 0.8 | 0.9 |
| Palmitoleic acid (16:1 n − 7) (%) | | 0.1 | 0.1 | 0.1 | 0.1 | 0.1 |
| Vaccenic Acid (18:1 n − 7) (%) | | 0.9 | 0.9 | 0.7 | 0.7 | 0.8 |
| **ω-9** | Total (%) | 17.1 | 6.0 | 25.8 | 15.4 | 12.9 |
| Oleic Acid (18:1 n-9) (%) | | 17.0 | 5.9 | 25.7 | 15.1 | 12.4 |
| Gondoic acid (20:1 n − 9) (%) | | 0.1 | 0.1 | 0.1 | 0.4 | 0.4 |
| **ω-3:ω-6** | Actual Ratio(n:1) | 3.837 | 3.084 | 0.002 | 0.249 | 0.272 |
| **Representative ratio** | | 11:3 | 3:1 | 1:614 | 1:4 | 2:7 |
| **Mineral composition** | | | | | | |
| Na (mg/kg) | | 1,073 | 0 | 0 | 0 | 0 |
| Mg (mg/kg) | | 4,425 | 3,525 | 3,493 | 5,359 | 3,296 |
| K (mg/kg) | | 7,482 | 7,679 | 7,930 | 9,747 | 7,367 |
| Ca (mg/kg) | | 1,388 | 5,689 | 1,289 | 349 | 982 |
| Cr (mg/kg) | | 2 | 2 | 1.5 | 2 | 1.3 |
| Mn (mg/kg) | | 23.3 | 30.6 | 31.5 | 68 | 91.5 |
| Fe (mg/kg) | | 143 | 145.4 | 80.6 | 196 | 196.3 |
| Co (mg/kg) | | 0.9 | 0.3 | 0.1 | 0.1 | 0.1 |

|  | Flax seed | Chia seed | Sunflower seed | Hulled hemp seed | Hemp flour |
|---|---|---|---|---|---|
|  | *Linum usitatissimum* | *Salvia hispanica* | *Helianthus annuus* | *Cannabis sativa* |  |
| Ni (mg/kg) | 1.2 | 1.8 | 5.2 | 53.2 | 0.6 |
| Cu (mg/kg) | 17.4 | 17.7 | 24.2 | 12.2 | 14.1 |
| Zn (mg/kg) | 43.7 | 52.0 | 66.1 | 81.7 | 53.0 |
| Se (mg/kg) | 0.9 | 0.9 | 0 | 1.8 | 1.0 |
| Mo (mg/kg) | 0.2 | 0.7 | 0.3 | 1.1 | 0.6 |
| I (mg/kg) | 274.4 | 267.6 | 403.7 | 459.4 | 314.7 |

## MTT cell viability assays

Both cell lines showed a similar response to each of the seed extracts (Figs. 2, 3, S2 and S3). Caco-2 and IEC-6 cell viability showed little change when treated with HHE or HFE when compared to the vehicle control (Figs. 2 and 3). FSE resulted in the largest increase in cell viability in both Caco-2 and IEC-6 cells (at the highest dose, 1 mg/mL, 163% and 195% of the vehicle control respectively) (Figs. 2 and 3).

## Protein and monosaccharide analyses

Analyses were performed to determine the protein and monosaccharide content of each crude aqueous seed flour extract. The protein analysis showed FSE had the highest levels of protein, followed by CSE when measured in the body temperature water extracts (Fig. 4). These levels differed slightly from the commercial label, which was based on the whole seed prior to extraction (Table 1). Flax, chia and hulled sunflower seed all originally had the same amount of protein per gram, however post-extraction, different levels were observed, with FSE having the highest protein content of 20.62 ± 1.28 (%w/w) and HSE having the lowest content of 7.60 ± 0.15 (%w/w) (Fig. 4). A monosaccharide analysis was performed which showed the presence of key polysaccharide building block monosaccharides, in varying concentrations between extracts (Fig. 4). CSE and FSE had the characteristic monosaccharides found in the mucilage fraction of each seed type. HSE, HFE and HHE extracts showed high levels of glucose but limited levels of the other monosaccharides quantified (Fig. 4).

## Mineral analysis of the aqueous extracts

Essential minerals were tested both pre- and post-extraction as this allowed for comparison of the availability of each mineral. The results showed high variability for each extract for the minerals measured (Fig. 5). For minerals such as potassium, chromium, selenium, iodine and copper, there was a slight increase in concentration from the flour to the extract from every seed. Sodium was only present in flax seed, with a three-fold increase in concentration in the extract. Hulled hemp and chia seed showed a similar level of magnesium in both the flour and corresponding extract, with the concentration increasing in the extract. The levels of magnesium were relatively similar in flax seed, hemp flour and hulled sunflower and did not change significantly in the extracts. Flax seed

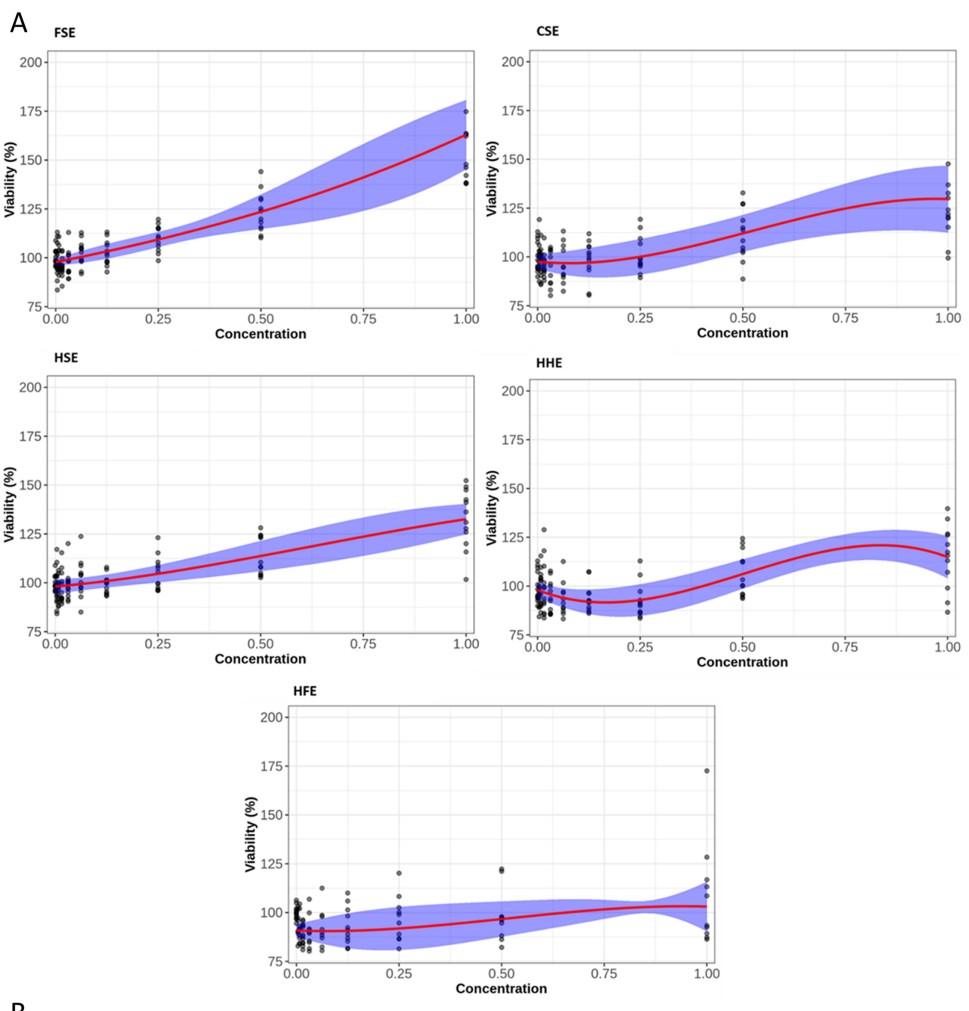

| | Flax Seed Extract | Chia Seed Extract | Hulled Sunflower Extract | Hulled Hemp Extract | Hemp Flour Extract |
|---|---|---|---|---|---|
| Control, 0 mg/mL | 100.00±0.00 | 100.00±0.00 | 100.00±0.00 | 100.00±0.00 | 100.00±0.00 |
| 0.625 mg/mL | **96.71±1.21 [a]**<br>**0.01 [b]** | **94.60±2.29**<br>**0.02** | **94.59±1.27**<br>**<0.001** | **89.71±4.79**<br>**0.03** | **88.90±2.04**<br>**<0.001** |
| 0.125 mg/mL | 100.47±2.29<br>0.84 | 99.27±2.80<br>0.79 | 97.26±2.80<br>0.33 | **90.35±3.71**<br>**0.01** | **92.81±3.08**<br>**0.01** |
| 0.25 mg/mL | **111.83±4.25**<br>**0.01** | 98.46±1.99<br>0.44 | 100.78±2.47<br>0.75 | **92.05±3.61**<br>**0.03** | 97.38±3.73<br>0.48 |
| 0.5 mg/mL | **128.79±9.37**<br>**0.01** | 108.27±4.74<br>0.08 | **111.02±3.89**<br>**0.01** | 99.89±7.66<br>0.99 | 99.89±3.24<br>0.97 |
| 1 mg/mL | **194.07±16.04**<br>**<0.001** | **116.45±6.39**<br>**0.01** | **138.57±13.83**<br>**0.01** | 106.45±5.73<br>0.26 | 101.27±3.89<br>0.74 |

**Figure 2** **Viability of Caco-2 cells after treatment with flax, chia, sunflower, and hemp crude aqueous extracts.** (A) Plotted dose-response models. (B) Table of mean viability estimated from the dose-response model. Bold values are statistically significant when compared to the vehicle control ($P < 0.05$). [a]Estimated mean ± SEM, [b]$P$-value of model contrasts comparing viability in a treatment to the control. Purple ribbon represents standard error, red line is estimated mean.

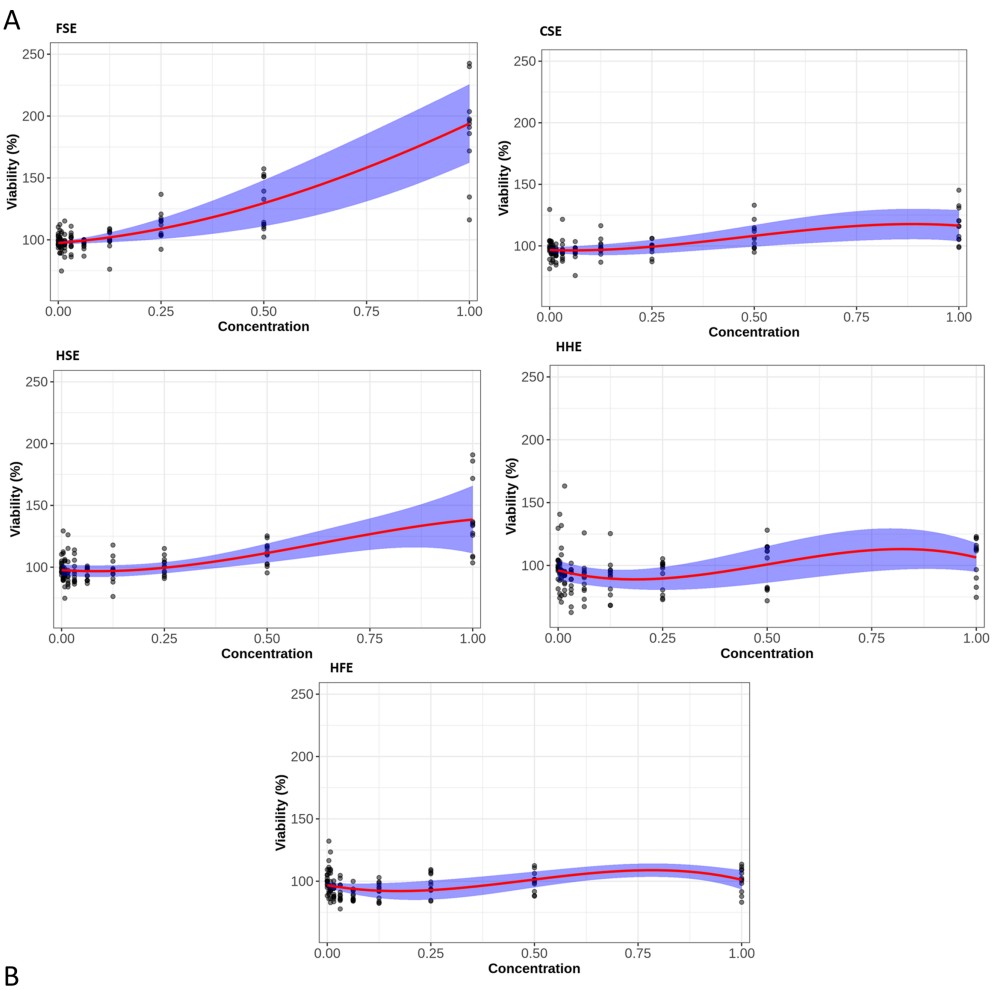

|  | Flax Seed Extract | Chia Seed Extract | Hulled Sunflower Extract | Hulled Hemp Extract | Hemp Flour Extract |
|---|---|---|---|---|---|
| Control, 0 mg/mL | 100.00±0.00 | 100.00±0.00 | 100.00±0.00 | 100.00±0.00 | 100.00±0.00 |
| 0.625 mg/mL | 102.37±1.47 [a] 0.11 [b] | 94.98±4.77 0.29 | 102.04±2.77 0.46 | **92.07±3.39 0.02** | **88.24±3.72 0.01** |
| 0.125 mg/mL | 102.35±1.98 0.24 | 98.49±3.14 0.63 | 99.34±1.44 0.64 | **93.82±2.98 0.04** | 91.58±4.42 0.06 |
| 0.25 mg/mL | **110.13±2.39 <0.001** | 101.13±3.93 0.77 | **104.70±2.23 0.04** | **92.34±3.81 0.04** | 93.59±5.38 0.23 |
| 0.5 mg/mL | **123.3±4.37 <0.001** | **111.57±5.14 0.02** | **113.74±3.98 <0.001** | 106.19±3.91 0.11 | 96.03±4.57 0.38 |
| 1 mg/mL | **163.00±8.98 <0.001** | **129.64±8.65 <0.001** | **132.57±3.92 <0.001** | **114.88±5.49 0.01** | 103.12±6.42 0.63 |

**Figure 3  Viability of IEC-6 cells after treatment with flax, chia, sunflower, and hemp crude aqueous extracts.** (A) Plotted dose-response models. (B) Table of mean viability estimated from the dose-response model. Bold values are statistically significant when compared to the vehicle control ($P < 0.05$). [a]Estimated mean ± SEM, [b]$P$-value of model contrasts comparing viability in a treatment to the control. Purple ribbon represents standard error, red line is estimated mean.

A

| | *Linum usitatissimum* | *Salvia hispanica* | *Helianthus annuus* | *Cannabis sativa* | |
|---|---|---|---|---|---|
| | Flax Seed Extract | Chia Seed Extract | Hulled Sunflower Extract | Hulled Hemp Extract | Hemp Flour Extract |
| Protein content (% w/w) | 20.62±1.28 | 17.82 ± 0.41 | 7.60±0.15 | 8.32±0.04 | 8.19±0.05 |
| **Monosaccharide Content** | | | | | |
| D-Mannose (% w/w) | nd | nd | nd | nd | nd |
| D-Ribose (% w/w) | nd | nd | 1.35 ± 0.03 | 1.41 ± 0.03 | nd |
| L-Rhamnose (% w/w) | 1.51 ± 0.13 | 0.44 ± 0.01 | nd | nd | nd |
| D-Glucuronic Acid (% w/w) | nd | 2.53 ± 0.01 | nd | nd | nd |
| D-Galacturonic Acid (% w/w) | 4.05 ± 0.14 | 2.30 ± 0.13 | nd | nd | 4.70 ± 0.25 |
| D-Glucose (% w/w) | 6.41 ± 0.30 | 6.46 ± 0.22 | 9.35 ± 0.10 | 6.45 ± 0.22 | 19.43 ± 0.10 |
| D-Galactose (% w/w) | 4.04 ± 0.32 | 4.11 ± 0.06 | 2.73 ± 0.04 | 0.82 ± 0.01 | 2.48 ± 0.00 |
| D-Xylose (% w/w) | 5.33 ± 0.60 | 5.04 ± 0.04 | nd | nd | 0.88 ± 0.01 |
| L-Arabinose (% w/w) | 1.67 ± 0.17 | 0.72 ± 0.01 | 0.19 ± 0.02 | 0.11 ± 0.01 | 0.50 ± 0.00 |
| L-Fucose (% w/w) | 0.74 ± 0.07 | nd | nd | nd | nd |
| | | | | | |
| **Mineral Content** | | | | | |
| Na (mg/kg) | 2969 | 0 | 87 | 3 | 509 |
| Mg (mg/kg) | 4101 | 7180 | 4434 | 3725 | 6988 |
| K (mg/kg) | 16318 | 32052 | 22205 | 18152 | 49942 |
| Ca (mg/kg) | 838 | 3395 | 903 | 321 | 2320 |
| Cr (mg/kg) | 3 | 2 | 2 | 2 | 2 |
| Mn (mg/kg) | 7 | 30 | 24 | 54 | 226 |
| Fe (mg/kg) | 141 | 111 | 149 | 122 | 110 |
| Co (mg/kg) | 2 | 2 | 0 | 0 | 0 |
| Ni (mg/kg) | 4 | 8 | 16 | 3 | 6 |
| Cu (mg/kg) | 37 | 18 | 76 | 20 | 45 |
| Zn (mg/kg) | 21 | 26 | 83 | 58 | 97 |
| Se (mg/kg) | 3 | 3 | 3 | 3 | 4 |
| Mo (mg/kg) | 0 | 1 | 1 | 3 | 3 |
| I (mg/kg) | 608 | 789 | 585 | 418 | 982 |

B

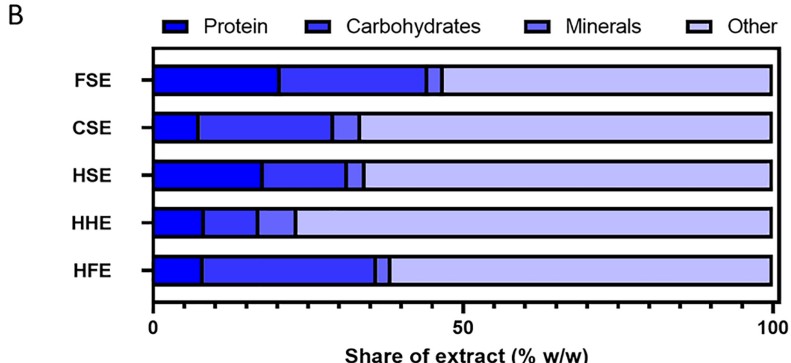

**Figure 4 Summary of aqueous extract composition.** (A) Protein concentration, aqueous extract monosaccharide and mineral profiles. (B) Stacked graph summarising the quantified components in the aqueous extracts.

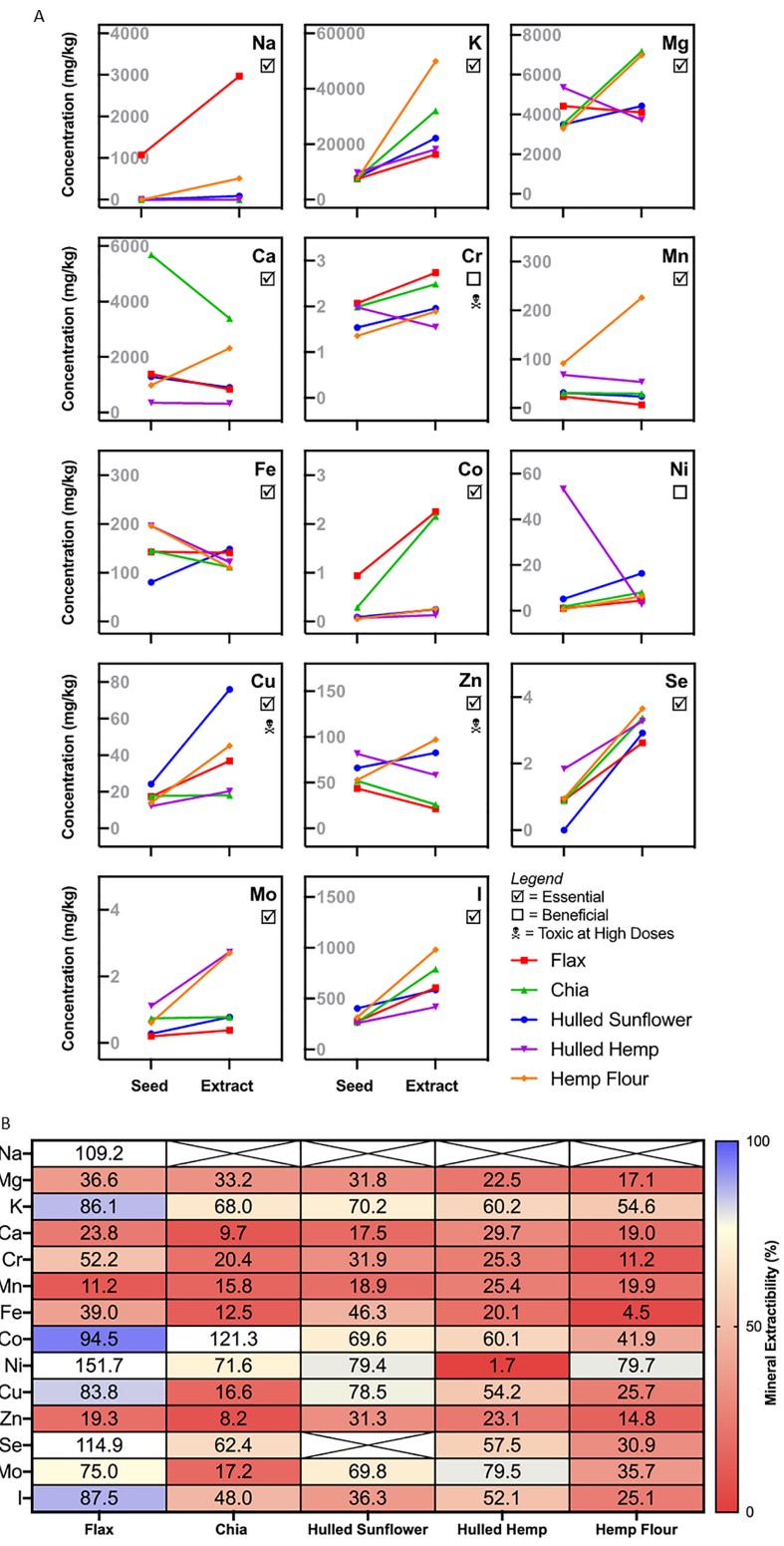

**Figure 5 Mineral profiling comparing aqueous extracts to their source flour.** (A) Change in absolute mineral concentration from source flour to aqueous extracts. (B) Extractability of minerals from sunflower, flax, chia, and hemp seed flour represented as the percentage from whole seed found in the aqueous extracts. Crossed values are unable to be calculated due to zero values (below quantitation limit).

showed consistent iron levels in both the flour and the extract, hulled sunflower showed a slight increase in the extract and chia seed, hulled hemp and hemp flour showed a slight decrease (Fig. 5). Chia seed had higher levels of calcium in the flour sample, but there was a decrease in the extract. Manganese showed a substantial increase in concentration in the extract for hemp flour only. Cobalt was enriched in the extracts from flax seed and chia seed compared to the initial flour sample. Nickel showed an initial high concentration in hulled hemp but was almost entirely absent in the extract.

### Extract composition

The protein, monosaccharide and mineral analyses were combined and compared for the total extracted material (Fig. 4). This figure shows that even with the combined protein, carbohydrates and minerals content quantified, there was still over 50% of the aqueous extract which was not categorised. The extract composition of FSE and HSE is predominantly protein, while for HFE it is carbohydrates. For CSE and HHE, more than 75% of the extract is currently uncharacterised.

### Antioxidant capacity of body temperature extracts

The antioxidant capacity of each of the seed extracts was determined using a ferric reducing antioxidant power (FRAP) assay (Fig. S1). HFE and HHE extracts had particularly low antioxidant capacity, while FSE and CSE showed a slight but significantly higher antioxidant capacity ($P < 0.05$). HSE showed the highest antioxidant capacity of all the extracts.

## DISCUSSION

Many seeds and plant products are declared to be superfoods, with multiple health benefits that often have little scientific evidence to support the claims. This study investigated the nutritional properties of both flour and flour extracts from several popular 'superfood' seeds, as well as the effects these extracts had on Caco-2 and IEC-6 cell viability. This provides us with evidence that these 'superfood' seed extracts may have gastrointestinal health benefits, derived from their ability to enhance cell viability of intestinal epithelial cells, as well as the beneficial nutritional profiles of the extracts. Seed extracts were made using a mild aqueous extraction method then properties such as mineral and carbohydrate content, protein content, and antioxidant capabilities were tested.

Techniques for producing botanical extracts are typically solvent-reliant (*e.g.*, hydroalcoholic) and often use high temperature/pressure to obtain high yields of often lowly-abundant bioactive compounds (*Santana & Macedo, 2019*; *Zimmer et al., 2021*). While these techniques are efficient, they likely over-extract and over-represent cytotoxic compounds in the extracts which may make them not ideal for assessing the health-promoting activities of botanical materials (*Alminger et al., 2014*; *Bohn et al., 2018*; *Minekus et al., 2014*). To produce a more biologically-relevant extract for such testing, we developed a simple, yet novel, aqueous extraction method performed at human body temperature (37 °C) to isolate water-extractable constituents from five 'superfoods' derived from seeds (Fig. 1). This method successfully produced crude aqueous extracts from flax,

chia, hulled sunflower and hulled hemp seeds as well as hemp flour, hereafter referred to as FSE, CSE, HSE, HHE, and HFE, respectively. This extraction method is not perfect and ideally future experiments would use an *in vitro* extraction method that more closely mimics the digestive system. This would likely give results more reflective of what would be obtained from ingestion of these seeds and flours as it includes gut digestion enzymes and multiple pH changes, providing a better indication of what the body is able to extract from ingested seeds (*Glahn et al., 1998*).

It is likely that the extraction conditions were not suitable for polysaccharide mobilisation from the cell walls of the different tissues of all seeds tested, as in flax and chia extracts, so monosaccharides characteristic of highly water-soluble mucilage were most of what was detected (*Lin, Daniel & Whistler, 1994*; *Naran, Chen & Carpita, 2008*). It is also possible that there are insoluble and acid hydrolysis-resistant carbohydrates present that are not quantifiable by the monosaccharide profiling method used here. Therefore, different methods could be tested to increase quantification recovery of polysaccharides such as by Saeman's hydrolysis (*Saeman, 1945*) or more sophisticated saccharide profiling could be employed like methylation linkage analysis (*Pettolino et al., 2012*).

To provide an indication of any health benefits/toxicity of the 'superfood' extracts, increasing doses were applied to IEC-6 rat small intestine and Caco-2 (Figs. 2 and 3) human large intestine epithelial cells grown *in vitro*, after which cell viability was measured (*Ponce de León-Rodríguez, Guyot & Laurent-Babot, 2019*). Using non-linear models, the relationship between extract dose and cell viability was studied, showing that even at the highest dose, 1 mg/mL, no extract showed significant cytotoxic effects. Several concentrations of flax, chia and sunflower extracts were found to increase the viability of IEC-6 and Caco-2 cells. FSE showed the greatest change with approximately 195% increase in viability at the highest concentration measured (1 mg/mL), followed by CSE and HSE, at approximately 140% and 115% respectively. The HHE and HFE maintained a consistent level of viability with no significant changes.

The MTT cell viability assay used here is essentially a measure of cellular metabolic activity, and thus the increases in cell viability observed are either a result of a decrease in apoptosis and/or an increase in proliferation. Extracts observed to enhance IEC-6 and Caco-2 cell viability likely contain several components that decrease apoptosis or increase cell proliferation including multiple beneficial micronutrients and antioxidant capabilities. FSE were enriched in sodium, chromium, cobalt, copper, selenium, and iodine and had a high protein content and antioxidant capabilities (Fig. 4). Studies have indicated that an increase in antioxidant activity can reduce apoptosis in healthy cells, which would result in a higher cell viability count (*Takahashi et al., 2013*; *Xu et al., 2012*).

A key factor preventing apoptosis in cells is the protection against oxidative stress, which can be prevented in the human body by antioxidants (*Aruoma, 1998*; *Maritim, Sanders & Watkins, 2003*). The FRAP assay used to measure antioxidant activity (Fig. S1) did show that FSE, CSE and HSE had the highest antioxidant capabilities,

however they did not directly match the increase in cell viability, as HSE had significantly higher antioxidant capabilities than both FSE and CSE. The HSE antioxidant activity was measured between those reported for *Ribes nigrum* (blackcurrant) buds and *Coffea Arabica* (arabica coffee) seeds (*Dudonne et al., 2009*). Although FRAP was the only method used in this study, other methods such as a DPPH (2,2-diphenyl-1-picryl-hydrazyl-hydrate) assay or an ABTS (2,2′-azino-bis(3-ethylbenzothiazoline-6-sulfonic acid) could also be used to measure different antioxidant capabilities (*Shah & Modi, 2015*). It is also possible that antioxidants combined with other factors could result in the increased cell viability observed (*Knight, 2000*). However, as oxidative stress was not specifically induced in this study this is difficult to include and it is likely that other mechanisms are also involved.

## CONCLUSIONS

The aim of this study was to determine whether superfood seeds have potential health benefits. To achieve this we developed a simple method for generating seed extracts with conditions that more closely mimics human physiology. We found an increase in cell viability of both the Caco-2 and IEC-6 cells with increasing amounts of flax, chia and hulled sunflower extracts. Whilst the extraction method used was beneficial for this study in decreasing the risk of alcohol-soluble cytotoxic compounds contaminating and degrading the extracts, future experiments could explore an *in vitro* extraction method, using gut digestion enzymes and multiple pH changes to continue to improve the biologically-relevant extraction of 'superfood' constituents.

Further research is needed to fully investigate the mechanisms by which some of the extracts increased cell viability including cell proliferation, oxidative stress, and/or inflammation pathway changes. In addition, other models can be used to investigate the potential health benefits of these superfood seeds and in particular whether gut health is improved. The microbiome could also be explored since this is crucial for optimal gut health. This research highlights that 'superfoods' may indeed have some of the health-promoting activities they are lauded for and the techniques used here, a simple aqueous extraction technique and the treatment of gastrointestinal cells *in vitro*, may provide simple methods to begin to provide evidence of clinical benefits.

## ACKNOWLEDGEMENTS

The authors want to express their gratitude to Sandy Khor and Anh Ngoc Hoang Nguyen for their assistance in cell culturing and monosaccharide analysis.

### Funding

This work was supported by funding from the University of Adelaide Women's Research Excellence Award (TBM). The funders had no role in study design, data collection and analysis, decision to publish, or preparation of the manuscript.

## Grant Disclosures

The following grant information was disclosed by the authors:
University of Adelaide Women's Research Excellence Award (TBM).

## Competing Interests

Rachel A. Burton is an Academic Editor for PeerJ.

## Author Contributions

- Jacqueline P. Barsby conceived and designed the experiments, performed the experiments, analyzed the data, prepared figures and/or tables, authored or reviewed drafts of the paper, and approved the final draft.
- James M. Cowley conceived and designed the experiments, analyzed the data, prepared figures and/or tables, authored or reviewed drafts of the paper, and approved the final draft.
- Shalem Y. Leemaqz analyzed the data, prepared figures and/or tables, authored or reviewed drafts of the paper, and approved the final draft.
- Jessica A. Grieger analyzed the data, authored or reviewed drafts of the paper, and approved the final draft.
- Daniel R. McKeating performed the experiments, analyzed the data, authored or reviewed drafts of the paper, and approved the final draft.
- Anthony V. Perkins analyzed the data, authored or reviewed drafts of the paper, and approved the final draft.
- Susan E. P. Bastian conceived and designed the experiments, analyzed the data, authored or reviewed drafts of the paper, and approved the final draft.
- Rachel A. Burton conceived and designed the experiments, analyzed the data, authored or reviewed drafts of the paper, and approved the final draft.
- Tina Bianco-Miotto conceived and designed the experiments, analyzed the data, authored or reviewed drafts of the paper, and approved the final draft.

## Data Availability

   All raw data is available in the Supplemental File.

## Supplemental Information

Supplemental information for this article can be found online at http://dx.doi.org/10.7717/peerj.12525#supplemental-information.

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
