# Peer review of "Nutritional properties of selected superfood extracts and their potential health benefits"

_PeerJ, doi:10.7717/peerj.12525_

## Round 0.1 · original submission · Major Revisions

Authors, please kindly attend to the concerns raised, and provide detailed information where requested. Looking forward to your revised manuscript. Thank you

Reviewer 1 ·

Basic reporting

no comment. understandable

Experimental design

understandable

Validity of the findings

need deeper scientific explanation

Additional comments

Thank you for your manuscript. its a nice result and bring a new "human and environment friendly".

here are some comments, I hope this will help to improve the quality of the manuscript.

General comments
It is mentioned that: there is a lack of explanation about the health benefits of those superfoods from the perspective of scientific study (in the abstract, and line 61-62). But, there are already several nutrition values of those mentioned materials in line 74-84. I would suggest rewriting the sentence in the abstract and line 61-62 to avoid misunderstanding.

The term “superfoods” is a term for food products available in the market. Are those materials claimed as superfoods, in the market where authors buy those products?

Abstract.
Methods: I would suggest to mention the analysis; chemical constituent for mineral, etc etc and biological activity such as antioxidant and cell viability”
Future studies, isn’t this too specific to mention against inflammation and oxidative stress? Because it can be studied in other non-communicable disease markers. I would suggest just “investigating their rules in regulating non-communicable diseases”. Or, it is mentioned in the previous sentence that extract composition and antioxidant composition are not directly related. So, this could be a suggestion for further study to explore the specific compounds which are related to the antioxidant properties and other biological properties.

Introduction
Line 61: the citation is not in the list of references. As far as I understand, superfood is not allowed to be used in several countries (at least in EU countries) unless they are proven to have health value. So, I am curious about this paper and I can't find it in the reference list.

Line 66: these superfoods are claimed to have a large range of health benefits. Who claimed this? Please provide more information about this. For example from paper, from market survey, or from authors observation?

Line 86-90: this part seems trying to argue with previous study to support the methods used in this study. I would suggest providing the citation in this statement. Or probably, it might be better just to mention that the mild aqueous extraction has never been applied in those materials, without making any negative speculation from the previous method. In addition to that, the positive value of mild aqueous extraction can be added to support this statement with citation.

Materials and methods
Line 103-105: the name of the company or the producers were mentioned. I am not sure what the regulation is if the authors need permission to put this name in the manuscript. Please check again and clarify.

Line 113: please provide the citation.

Line 117: please convert the rpm to g unit.

Line 170: the “.” after the citation.

Line 178: what about the replication in the analysis?it is not mentioned in the manuscript.

Results
I would suggest to provide the level of significant difference according to the post-hoc test, whether 0.05 or 0.01.

Discussion
Line 280-284: the previous technique is efficient, but it may have an impact on health promoting activities. Please provide the citation and more detailed explanation. How much was the yield from the previous extraction compared to this study.
I would suggest, to provide several techniques which have been applied as comparison to the novel technique.

Line 304-313: could authors please provide some literature for the toxicity of solvent-based extracts to make a better understanding of this novel method?

Line 314-335: please identify or provide the literature, about the specific biological compounds in those materials which might be responsible for the antioxidant activity and other biological activity.

Conclusions
Line 334-343: the citation is not important in conclusion
Line 334-343: please remove this part at the end of the conclusion
I would suggest, the novel extraction technique must be stressed in conclusion and what are the benefits from this finding, followed by the possible study to improve this technique in near future.

References: please check again to include all the citations in the text to the references list.

·

Basic reporting

The author(s) should show if there is clear difference(s) between super-food and functional food as shown in LINE 52 - 58. Or can the names be used interchangeably?

Experimental design

The fatty acid composition protocol is missing (LINE 189).

while LINE 261 should read In vitro......

Validity of the findings

No comment

Additional comments

The manuscript is well written in a simple and constructive language, with minor modification as shown in the individual sections.

Reviewer 3 ·

Basic reporting

Authors report a simple aqueous extraction technique of specific superfoods and the treatment of gastrointestinal cells in vitro, trying to provide simple methods to provide evidence of their clinical benefits.
Since other compounds are higher than compounds studied it would be interesting to focus more on antioxidant compounds present in these superfoods. Results are not clearly support hypothesis of study.

Please check my comments below on experimental work, result expression and literature cited.

Experimental design

L 11, for Lipid content and fatty acid profiles of flours please give a brief experimental description
For crude aqueous body temperature extracts, authors could have used acid since in the stomach acid conditions occur. Is the method used in house? if not please add reference/s and explain why specific amounts of flour and water were used (4g flour to 200ml water)..
L. 121-121 did authors prepare 5mg/ml in RO water for for compositional analysis as well. What RO stands for? why not use RO instead of deionized water for extraction? At L. 136-137 authors say that Extracts (5 mg/mL) were diluted in DMEM but at L. 120 in PBS, there is a conflict here, please explain.
Please add reference/s for IEC-6 and Caco-2 Cell Culture section experiment.
monosaccharides analysis is not clearly presented, how 1-phenyl-3-methyl-5-pyrazoline (PMP) derivatives were prepared? Quantification of results how it was expressed? a typical HPLC chromatogram should be added.
A brief description of ICP-MS should be presented. Validation of method is not presented, what are the LOD and LOQ of each mineral?

Validity of the findings

Results of monosaccharides analysis are presented as D- and L- however at M&M thay are not mentioned the same way, also results are %w/w why not in concentrations of initial amount of flour used, please revise
For mineral content std deviation should be added to results.
Fig 2 and 3: error bars are not visible to check significant differences, also there are a and b for statistical analysis for flaxseed extract, why only for one value?
Significant digits should be checked for results at figure 3 for 1mg/ml flax seed extract

Additional comments

Concerning references they need to be rechecked for example, Arai 2002, Proestos 2018, Goyal et al., 2014; Ixtaina et al., 2011, Callaway, 2004, Mohamed et al.,2020 and many more are missing from Reference list even though they are present within text, please revise.
why authors used Ferric Reducing Antioxidant Power (FRAP) Assay? this is usually for biological samples and when antioxidants like phenolics are studied, please explain.

---

## Round 0.2 · Minor Revisions

Please authors, the reviewers have commented on work, and request a few minor touches here and there. In addition, the editor encourages authors to address the following:

a) Kindly amend the title to read: "Nutritional properties of selected superfood extracts and their potential health benefits" to be consistent with the objective of this work.
b) In the introduction, more information is required regards extraction methods that have been used by previous studies of superfood seeds. So before Line 90, authors should provide a new paragraph, with the following information:
-Which (array of) extraction methods have been used by previous studies of superfood seeds (particularly those mentioned earlier in the introduction), and why were they used
-Which method posed greatest or least challenge, and how did previous authors overcome them?
-Which extraction method provided best results, and which ones appeared favourable? The essence here is to help authors justify why the one that has been used for this current study, is being used
The above three points should form the new paragraph before Line 90
The sentence starting: "The objective of .... (Line 90) should be the start of the last paragraph of the introduction
c) In the methods, authors are encouraged to create a new subsection captioned 'Schematic overview of the experimental program", which should comprise 3-4 sentences, and be supported with a flow diagram - this should be your Figure 1 (i.e. diagram with arrows showing : Purchase of commercially available samples> Blade mill grinding of samples >Nutritional information from commercial packaging
-crude aqueous extract preparation>Crude aqueous extract preparation>.....> analytical determinations >>>> (then sub-arrows directed to the individual analysis)
Please authors, Lines 103 to 115 must be expanded. Materials must strictly be materials only. Provide new sub-sections captioned:
- Blade mill grinding of samples
- Nutritional information from commercial packaging
-Crude aqueous extract preparation
etc etc
Please, provide more details of steps that were actually taken to conduct these.
For instance, blade mill grinding, what quantity was put, after grinding, how were ground samples kept/stored?
For instance, Nutritional information from commercial packaging, why did you do this? What is the basis for it, is it for the purpose of comparisons?
Lines 112 to 115, are these not analytical determinations? Please move them down to the analytical determinations ok
The current Figure 1, will then become Figure 2.
Why was crude aqueous body temperature generated? Please state why, and support it with reference, Line 116, and Crude Aqueous Body Temperature Seed Extracts entire procedure, is it your own? If it is not, please indicate the source, and if it is modified , state so

Why did you do IEC-6 and Caco-2 Cell Culture? Please, support it with a reference, is it your own method, if it is from a source, state it

The same applies to Cell Viability Assay, you modifed it from Kumar et al. 2018, isn't it? Please state so

Please go through the rest of others, and check them appropriately
Statistical analysis, what was your level of statistical significance? Please, state it . Which statistical analyses were performed using R version 3.6? Please, be specific, so that we know what GenStat 15th Edition was used for, and what R version 3.6, was used for? so lines 195-196, should clearly indicate which specific ones were run using
R version 3.6,

c) Results
Please, kindly remove all references cited in the results section, results are strictly results, example, Line 242-243, Line 266-268,
Please, all results of statistical significance must be supported with their exact P-values. Please make sure you provide these. Also, provide the R-sq adj values, since you used R version 3.6.

d) Discussion
Please, go through the entire discussion, kindly provide in bracket (Refer to Table ?) or (Refer to Figure?) in all places where results of either Table or Figure is being presented. That is to say, all the Table or Figure mentioned in the result must be captured in the discussion. This is to guide readers to understand what aspects of the results are being discussed ok

e) Conclusion
Please, revise the conclusion using the following:
- reiterate what the hypothesis/rationale of this work, and why it was embarked upon
-why the extraction method used was used, and why it is of benefit to this study
-what were the key results and why are they considered key, what are the implications of the results
These above three points should be first paragraph of conclusion
Then, Lines 347 to 355 should be expanded a bit, contextualised better, authors should apply their discretion to achieve this

Please, authors are encouraged to address every single point raised here. It will help to strengthen this very good study.

Look forward to your revised manuscript.

Thank you

Reviewer 1 ·

Basic reporting

Well described

Experimental design

Well described

Validity of the findings

Well described

Additional comments

All comments have been addressed well, EXCEPT 1 more in conclusion: the citation is unnecessary in conclusion, which has not removed yet. even thought its mentioned in the rebuttal letter the revision has been made, in fact no revision is made in conclusion part.

·

Basic reporting

The report is well written in simple language that would enable even non experts to understand the work. Though, not without some common errors as highlighted.

LINE 27 help with or should be deleted

LINE 63 'both' and 'a' should be deleted

Experimental design

The experimental designs are appropriate.

Validity of the findings

The findings are valid within the concept design.

Reviewer 3 ·

Basic reporting

Article is written in clear, understandable English language
Authors added references that were missing.
The structure, the figures and Tables are corrected according to my comments and suggestions.

Experimental design

All queries concerning experimental procedures, materials and methods and statistical analysis were adequately replied. Response is accepted.

Validity of the findings

Results are clear and well presented. Response from authors regarding result expression is satisfactory and accepted. Conclusions are supported by results, well stated. Authors provide all data replying to my comments.

Additional comments

Article is written in clear, understandable English language
Authors added references that were missing.
The structure, the figures and Tables are corrected according to my comments and suggestions.

---

## Round 0.3 · accepted · Accept

Thank you authors for revising your work. It is now acceptable for publication. Thank you for finding PeerJ as your journal of choice, and making the very best of the peer-review process, which improved the quality of this scholarly piece of work. Looking forward to your future scholarly contributions. Thank you very much
Congratulations and very best wishes